

# A high-resolution regional emission inventory of atmospheric mercury and its comparison with multi-scale inventories: a case study of Jiangsu, China

Hui Zhong[1], Yu Zhao[1, 2*], Marilena Muntean[3], Lei Zhang[4], Jie Zhang[2, 5]

1. State Key Laboratory of Pollution Control & Resource Reuse and School of the Environment, Nanjing University, 163 Xianlin Ave., Nanjing, Jiangsu 210023, China

2. Jiangsu Collaborative Innovation Center of Atmospheric Environment and Equipment Technology (CICAEET), Nanjing University of Information Science & Technology, Jiangsu 210044, China

3. European Commission, Joint Research Centre, Institute for Environment and Sustainability, Air and Climate Unit, Via E. Fermi, Ispra, Italy

4. University of Washington-Bothell, 18115 Campus Way NE, Bothell, WA 98011, U.S.A.

5. Jiangsu Provincial Academy of Environmental Science, 176 North Jiangdong Rd., Nanjing, Jiangsu 210036, China

* Corresponding author: Phone: 86-25-89680650; email: yuzhao@nju.edu.cn



**ABSTRACT**

2       A better understanding of the discrepancies in multi-scale inventories could give

an insight on their approaches and limitations, and provide indications for further
improvements; international, national and plant-by-plant data sources are primarily
obtained to compile those inventories. In this study we develop a high-resolution
inventory of Hg emissions at $0.05^o \times 0.05^o$ for Jiangsu China using a bottom-up
approach and then compare the results with available global/national inventories. With
detailed information on individual sources and the updated emission factors from field
measurements incorporated, the annual Hg emissions of anthropogenic origin in
Jiangsu 2010 are estimated at 39 105 kg, of which 51%, 47% and 2% were released as
$Hg^0$, $Hg^{2+}$, and $Hg^P$, respectively. This provincial inventory is thoroughly compared to
the downscaled results from three national inventories (NJU, THU and BNU) and two
global inventories (AMAP/UNEP and EDGARv4.tox2). Attributed to varied methods
and data sources, clear information gaps exist in multi-scale inventories, leading to
differences in the emission levels, speciation and spatial distributions of atmospheric
Hg. The total emissions in the provincial inventory are the largest, i.e., 28%, 7%, 19%,
22%, and 70% higher than NJU, THU, BNU, AMAP/UNEP, and EDGARv4.tox2,
respectively. For major sectors including power generation, cement, iron & steel and
other coal combustion, the Hg contents ($HgC$) in coals/raw materials, abatement rates
of air pollution control devices (APCD) and activity levels are identified as the crucial
parameters responsible for the differences in estimated emissions between inventories.
Regarding speciated emissions, larger fraction of $Hg^{2+}$ is found in the provincial
inventory than national and global inventories, resulting mainly from the results by
the most recent domestic studies in which enhanced $Hg^{2+}$ were measured for cement
and iron & steel plants. Inconsistent information of big power and industrial plants is
the main source of differences in spatial distribution of emissions between the
provincial and other inventories, particularly in southern and northwestern Jiangsu
where intensive coal combustion and industry are located. Quantified with
Monte-Carlo simulation, uncertainties of provincial Hg emissions are smaller than
those of NJU national inventory, resulting mainly from the more accurate activity data
of individual plants and the reduced uncertainties of $HgC$ in coals/raw materials.





# 1 INTRODUCTION

Mercury (Hg), known as a global pollutant, has received increasing attention for
its toxicity and long-range transport. Identified as the most significant release into the
environment (Pirrone and Mason, 2009; AMAP and UNEP, 2013), atmospheric Hg is
analytically defined as: gaseous elemental Hg (GEM, $Hg^0$) that has longest lifetime
and transport distance, and reactive gaseous mercury (RGM, $Hg^{2+}$) and particle-bound
mercury (PBM, $Hg^p$) that are more affected by local sources. Improved estimates in
emissions of speciated atmospheric Hg are believed to be essential for better
understanding the global transport, chemical behaviors and mass balance of Hg.
Due mainly to the fast growth in economy and intensive use of fossil fuels, China
has been indicated as the highest ranking nation in anthropogenic Hg emissions (Fu et
al., 2012; Pacyna et al., 2010; Pirrone et al., 2010). Emissions of speciated
atmospheric Hg of anthropogenic origin in China have been estimated at both global
and national scales. For example, AMAP/UNEP (2013) and Muntean et al. (2014)
developed global Hg inventories, with national-specific emissions reported for China
for 2010 and from 1970 to 2008, respectively. At national scale, Hg emissions have
been estimated based on more detailed provincial information on energy consumption
and industrial production. Zhang et al. (2015), Zhao et al. (2015a) and Tian et al.
(2015) evaluated the inter-annual trends in emissions for 2000-2010, 2005-2012, and
1949-2012, respectively, to explore the benefits of air pollution control polices,
particularly for recent years.
There are considerable information gaps between multi-scale inventories,
attributed mainly to the data of different sources and levels of details. For coal-fired
power plants (CPP), as an example, the global inventories by AMAP/UNEP (2013)
and Muntean et al. (2014) obtained the national coal consumption from the
International Energy Agency (IEA), and they acquired the information of control
technologies from the "national comments" by selected experts and World Electric
Power Plants database (WEPP), respectively. In the national inventory by Zhang et al.
(2015) and Tian et al. (2015), coal consumption of CPP by province was derived from
official energy statistics, and the penetrations of flue gas desulfurization (FGD)
systems were assumed at provincial level. Zhao et al. (2015a) further analyzed the
activity data and emission control levels plant by plant using a "unit-based" database
of power sector. Although data of varied sources and levels of details result in



discrepancies between inventories, those discrepancies and the underlying reasons
have not been thoroughly analyzed in previous studies, leading to big uncertainty in
Hg emission estimation.
Existing global and national inventories could hardly provide satisfying estimates
in speciated Hg emissions or well capture the spatial distribution of emissions at
regional/local scales, attributed mainly to relatively weak investigation on individual
sources. When they are used in chemistry transport model (CTM), downscaled
inventories at global/national scales would possibly bias the simulation at smaller
scales. Improvement in emission estimation at local scale, particularly for the large
point sources is thus crucial for better understanding the atmospheric processes of Hg
(Lin et al., 2010; Wang et al., 2014; Zhu et al., 2015). While local information based
on sufficient surveys is proven to have advantages in improving the emission
estimates for given pollutants like $NO_X$ and $PM_{10}$ (Zhao et al., 2015b; Timmermans et
al., 2013), there are currently very few studies focusing on Hg at regional/local scales,
and the differences of multi-scale inventories remain unclear.
In this work, therefore, we select Jiangsu, one of the most developed provinces
with serious air pollution in China, as study area. Firstly, we develop a high-resolution
Hg emission inventory of anthropogenic origin for 2010, based on comprehensive
review of field measurements and detailed information on emission sources. That
provincial inventory is then compared to selected global and national inventories with
a thorough analysis on data and methods of multi-scale inventories. Discrepancies in
emission levels, speciation, and spatial distributions are evaluated and the underlying
sources of the discrepancies are figured out. Finally, the uncertainty of the provincial
emission inventory is quantified and the key parameters contributing to the
uncertainty are identified. The results provide an insight on the effects of varied
approaches and data on development of Hg emission inventory, and indicate the
limitations of current studies and the orientations for further improvement on emission
estimation at regional/local scales.

**2 DATA AND METHODS**

**2.1 Data sources of multi-scale inventories**
As shown in Figure S1 in the supplement, Jiangsu province (30°45′ N-35°20′ N,
116°18′ E-121°57′ E) is located in Yangtze River Delta in eastern China and covers 13



cities. The Hg emissions of Jiangsu are obtained from two approaches: downscaled
from global/national inventories, and estimated using a bottom-up method with
information of local sources incorporated.
In global/national inventories, Hg emissions were first calculated by sector based
on activity data and emission factors that were obtained or assumed at global, national
or provincial level, and were then downscaled to regional domain with finer spatial
resolution. Various methods and data were adopted in multi-scale inventories to
estimate Hg emissions for different sectors, as summarized briefly in Table S1 in the
supplement. Three national inventories were developed by Nanjing University (NJU,
Zhao et al., 2015a), Beijing Normal University (BNU, Tian et al., 2015), and Tsinghua
University (THU, Zhang et al., 2015), with major activity data at provincial level
obtained from Chinese national official statistics. Compared to NJU and BNU
inventories that applied deterministic parameters relevant to emission factors, THU
developed a model with probabilistic technology-based emission factors to calculate
the emissions. Based on international activity statistics at national level, two global
inventories for 2010 were developed by the joint expert group of Arctic Monitoring
and Assessment Programme and United Nations Environment Programme
(AMAP/UNEP, 2013), and Emission Database for Global Atmospheric Research
(EDGARv4.tox2, unpublished). AMAP/UNEP inventory developed a new system for
estimating emissions from main sectors based on a mass-balance approach with data
on unabated emission factors and emission reduction technology employed in
different countries. EDGARv4.tox2 inventory calculated the emissions for all the
countries by primarily applying emission factors from EEA (2009) and USEPA (2012),
combined with regional technology-specific information of emission abatement
measures.

**2.2 Development of the provincial inventory**
In contrast to the downscaling approach, a bottom-up method is further applied,
in which the emissions are first calculated plant by plant based on information of
individual sources and then aggregated at provincial level. We mention the inventory
as bottom-up or provincial inventory hereinafter. Information for individual sources
are thoroughly collected from Pollution Source Census (PSC, internal data from
Environmental Protection Agency of Jiangsu Province), including combustion
technology, fuel quality and air pollutant control devices. According to the availability



of data, anthropogenic sources are classified into three main categories. Category 1
includes coal-fired power plants (CPP), iron & steel plants (ISP), cement production
(CEM) and other industrial coal combustion (OIB). Note that the emissions from coal
combustion in cement production are not included in CEM but in OIB, following
most other inventories involved in this work for easier comparison. The information
on geographic location, activity levels (consumption of energy or raw materials) and
penetration of air pollution control devices (APCDs) is compiled plant by plant from
Pollution Source Census, with an exception that the technology employed in CEM are
obtained from CCA (2011). Category 2 includes nonferrous metal smelting (NMS),
aluminum production (AP), municipal solid waste incineration (MSWI) and
intentional use sector (IUS: thermometer, fluorescent lamp, battery and polyvinyl
chloride polymer production). Geographic location information for those sources is
obtained from Pollution Source Census, while other activity data come from official
statistics at provincial level. Category 3 includes emission sources that are not
contained in Pollution Source Census: residential & commercial coal combustion
(RCC), oil & gas combustion (O&G), biofuel use/biomass open burning (BIO), rural
solid waste incineration (RSWI) and human cremation (HC). They are defined as area
sources, and the data sources for them are discussed later in this section.
In general, annual emissions of total and speciated Hg are calculated using Eq. (1)
and (2), respectively:

$$E = \sum_n AL_n \times EF_n \tag{1}$$

$$E_s = \sum_n AL_n \times EF_n \times F_{n,s} \tag{2}$$

where $E$ is the Hg emission; $AL$ is the activity levels (fuel consumption or industrial
production); $EF$ is the combined emission factor (emissions per unit of activity level);
$F$ is the mass fraction of given Hg speciation; $n$ and $s$ represent emission source type
and Hg speciation ($Hg^0$, $Hg^{2+}$ or $Hg^p$).
For CPP/OIB and CEM, Eq. (1) can be revised to Eq. (3) and (4) respectively,
with detailed fuel and technology information of individual sources incorporated:

$$E_{CPP/OIB} = \sum_t \sum_i \sum_k AL_i \times HgC_k \times RR_t \times (1 - RE_t) \tag{3}$$

$$E_{CEM} = \sum_t \sum_i (AL_{Limstone} \times HgC_{Limstone} + AL_{Other,i} \times HgC_{Other}) \times (1 - RE_t) \tag{4}$$

where $HgC$ is the Hg content of coal consumed in Jiangsu, calculated based on





measured Hg contents of coal mines across the country and an inter-provincial flow
model of coal transport (Zhang et al., 2015); $HgC_{Limestone}$ and $HgC_{Other}$ represent Hg
contents of limestone and other raw materials (e.g. malmstone and iron powder) in
cement production, respectively; $RR$ is the Hg release ratios from combustors; $RE$ is
Hg removal efficiency of APCDs; $AL_{Limestone}$ and $AL_{Other}$ represent the consumption of
limestone and other raw materials in CEM, respectively; $i$ and $k$ represent individual
point source and coal type, respectively; $t$ represent APCD type including wet
scrubber (WET), cyclone (CYC), fabric filter (FF), electrostatic precipitator (ESP),
FGD and selective catalyst reduction (SCR) systems for CPP, and dry-process
precalciner technology with dust recycling (DPT+DR), shaft kiln technology (SKT)
and rotary kiln technology (RKT) with ESP or FF for CEM. Note the $AL$ for
individual CEM plant is calculated based on the clinker and cement production when
the information on limestone or other raw materials is missing in PSC.

For ISP, Eq. (1) could be revised to Eq. (5):

$$E_{ISP} = \sum_i (AL_{steel,i} + AL_{iron,i} \times R) \times EF_{steel} \tag{5}$$

where $AL_{steel}$ and $AL_{iron}$ represent crude steel and pig iron production in ISP,
respectively; $R$ is the liquid steel to hot metal ratio provided by BREF (2012),
converting the production of pig iron to crude steel equivalent; $EF_{steel}$ is the Hg
emission factor applied to steel making, obtained from recent domestic tests by Wang
et al. (2016).

Activity data for NMS, AP, MSWI, RCC and O&G are derived from national

statistics (NMIA, 2011; NSB, 2011a; 2011b), while Hg consumption in IUS are
estimated based on the internal industry reports. The biofuel use is obtained from the
investigation by Ministry of Agriculture (C. Chen et al., 2013). The biomass
combusted in open fields is originally calculated as a product of grain production,
waste-to-grain ratio, and the percentage of residual material burned in the field, as
described in Zhao et al. (2011, 2012). The rural municipal waste burned are calculated
as a product of rural population, the average waste per capita, and the ratios of waste
that is burned (Yao et al., 2009). Other information including control efficiencies of
APCDs, speciation profiles and emission factors inherited from previous studies is
summarized in Table S2-S4 in the supplement.

Regarding the spatial pattern of emissions, the study domain is divided into 4212

grid cells with a resolution at 0.05˚×0.05˚. For Categories 1 and 2, emissions are





directly allocated into corresponding grid cells according to the locations of individual
sources. As considerable errors of plant locations were unexpectedly found in PSC,
the geographic location for point sources with emissions more than 15 kg have been
corrected by Google Map. As a result, totally 900 plants are relocated, accounting for
14% of all the point sources. For Category 3, emissions are allocated according to the
population density in urban areas (RCC) and that in rural areas (BIO and RSWI).

**2.3 Sensitivity and uncertainty analysis**
For better understanding the sources of discrepancies between inventories, a
comprehensive sensitivity analysis is conducted to quantify the differences between
selected parameters used in multi-scale inventories and the subsequent changes in
emission estimation for Category 1 sources. The relatively change ($RC$) of given
parameter ($j$) in global/national inventories compared to those in the provincial
bottom-up inventory, and the changes in Hg emissions for selected source ($n$) when
the value of given parameter in the bottom-up inventory is replaced by that in
global/national inventories ($E_{diff,n}$), can be calculated using Eqs. (6) and (7),
respectively:

$$RC_j = (VO_j - VB_j)/VB_j \tag{6}$$

$$E_{diff,n} = EO_n - EB_n \tag{7}$$

where $VB$ is the value of parameters in bottom-up inventory; $VO$ is the value of
parameters in other national/global inventories; $EB$ is Hg emissions for given sector in
bottom-up inventory; $EO$ is Hg emissions for given sector when the values of
parameters in bottom-up inventory are replaced by those in other global/national
inventories; $j$ and $n$ represent given parameter and source type, respectively.
In particular, a new parameter, total abatement rate ($TA$), is defined for the
sensitivity analysis, combining the effect of the penetrations of APCDs and their
removal efficiencies on emission abatement:

$$TA = \sum_t AR_t \times RE_t \tag{8}$$

where $t$ represents APCD type; $AR$ and $RE$ are the application rate and Hg removal
efficiency, with detailed information provided in Table S5 in the supplement.
The uncertainties of speciated Hg emissions at provincial level are quantified
using a Monte-Carlo framework (Zhao et al., 2011). Given the relatively accurate data





reported in PSC, the probability distributions of activity levels for individual plants of
CPP, OIB, ISP and CEM are defined as normal distributions with the relative standard
deviations (RSD) set at 10%, 20%, 20% and 20% respectively. As summarized in
Table S6 and Table S7 in the supplement, a database for Hg emission factors/related
parameters by sector and speciation for main sources are established for China, with
the uncertainty analyzed and presented by probability distribution function (PDF).
The PDFs of Hg contents in coal mines by province are obtained from Zhang et al.
(2015). For Hg content in limestone ($HgC_{Limestone}$), a lognormal distribution is
generated with bootstrap simulation based on 17 field tests by Yang (2014), as shown
in Figure S2 in the supplement. For the rest parameters, a comprehensive analysis of
uncertainties were conducted with the results of field measurements available fully
incorporated as described in Zhao et al. (2015a). Ten thousand simulations are
performed to estimate the uncertainties of emissions, and the parameters that are most
significant in determination of the uncertainties are identified by source type
according to the rank of their contributions to variance.

**3 RESULTS AND DISCUSSIONS**
**3.1 Emission estimation and comparison by sector**
**3.1.1 The total Hg emissions from multi-scale inventories**

Table 1 provides the Hg emissions by sector and species for Jiangsu 2010

estimated from the bottom-up approach. The provincial total Hg emissions of
anthropogenic origin are calculated at 39 105 kg, of which 51% released as $Hg^0$, 47%
as $Hg^{2+}$, and 2% as $Hg^P$. In general, Categories 1, 2 and 3 account for 90%, 4% and
6% of the total emissions, respectively. CPP and CEM are the biggest
contributors to the total Hg ($Hg^T$) emissions. For $Hg^0$, $Hg^{2+}$, and $Hg^P$, the sectors with
the largest emissions are CPP, CEM, and OIB respectively.

To better understand the discrepancies and their sources between various studies,

the emissions from multi-scale inventories are also summarized in Table 1 for
comparison. Among all the inventories, the total emissions in the provincial inventory
are the largest, i.e., 28%, 7%, 19%, 22%, and 70% higher than NJU, THU, BNU,
AMAP/UNEP, and EDGARv4.tox2, respectively. The elevated Hg emissions
compared to previous studies could be supported by modeling and observation work





to some extent. Based on the chemistry transport modeling using GEOS-Chem (Wang
et al., 2014), or correlation slopes with certain tracers (CO, $CO_2$ and $CH_4$) from
ground observation (Fu et al., 2015), underestimation was suggested for the regional
Hg emissions of anthropogenic origin in China.
Direct comparison of emissions between inventories is unavailable for every
sector, as the definition of source categories is not fully consistent with each other.
Therefore, necessary assumption and modification are made on source classification
for global inventories. In Table 1, CPP, OIB and RCC for EDGARv4.tox2 actually
represent the emissions for all the fossil fuel types, and they are 1316, 5342 lower and
986 kg higher than our estimation from coal combustion, respectively. For
AMAP/UNEP, the emissions from regrouped stationary combustion (industrial
sources excluded), industry, and intentional use and product waste associated sources
(see Table 1 for the detailed definition) are respectively 3382, 2032 higher and 3118
kg lower than our estimation with bottom-up method. Figure 1 shows the ratios of the
estimated Hg emissions in national/global inventories to those in the provincial
inventory by source. The CPP emissions are relatively close to each other, but larger
differences exist in some other sources. The estimates for CEM and ISP in provincial
inventory are much higher than NJU, BNU and EDGARv4.tox2 inventories, while
those for NMS are extremely smaller. The reasons for those differences are discussed
and analyzed in details in Sections 3.1.2 and 3.1.3.
**3.1.2 Sensitivity analysis for Category 1 sources**
Figure 2 (a) and (b) represents the relative changes in given parameters between
the provincial and other inventories, and the subsequent differences in Hg emissions
for Category 1 sources, using Eqs. (6) and (7), respectively. For CPP, the differences
between provincial and national/global inventories are mainly determined by *AL*, *HgC*,
*TA*, and *IEF*, as indicated by the calculation methods summarized in Table S1.
(Instead of analyzing *HgC* and *RR* separately, integrated input emission factors (*IEF*)
were applied in AMAP/UNEP and EDGARv4.tox2.) For activity level (*AL*), the
coal consumption data are collected and compiled plant by plant in the provincial
inventory, while they were obtained from Chinese official statistics (NSB, 2011b) in
national inventories. As a result, the coal consumptions in NJU and THU inventories
are 17% and 6% smaller than our provincial inventory, resulting in 1968 and 760 kg
reduction in Hg emission estimate, respectively.





In national and provincial inventories, as mentioned in Section 2, the Hg contents
in the raw coal ($HgC_{raw}$) consumed by province are estimated using an
inter-provincial flow matrix for coal transport based on the results of field
measurements on Hg contents for given coal mines (Tian et al., 2010; Tian et al., 2014;
Zhang et al., 2012). The $HgC_{raw}$ for Jiangsu in THU and our provincial inventory
come from Zhang et al. (2012), who merged the results of two comprehensive
measurement studies on $HgC_{raw}$ for coal mines across China after 2000, by
themselves and USGS (2004), and the average value is calculated at 0.2 g/t-coal. NJU
inventory adopted the $HgC_{raw}$ of 0.169 g/t-coal from Tian et al. (2010), while BNU
inventory determined $HgC_{raw}$ at 0.25 g/t-coal with a bootstrap simulation based on a
thorough investigation on published data (Tian et al., 2014). $HgC_{raw}$ in NJU and BNU
inventories are 15% smaller and 25% higher than that in provincial inventory, leading
to differences of 1746 and 2816 kg in Hg emissions, respectively. Given the large
differences in $HgC_{raw}$ between countries, global inventories applied national specific
IEF based on the domestic tests (UNEP, 2011b; Wang et al., 2010). The IEFs for
China applied in AMAP/UNEP and EDGARv4.tox2, without considering the
regional differences in $HgC_{raw}$, are 26% and 28% lower than that in provincial
inventory (recalculated with $HgC_{raw}$ and $RR$). As regional $HgC_{raw}$ differs a lot from
the national average and could be largely influenced by the data selected, big
discrepancy might exist when national value is applied in regional inventory, and
more regional-specific measurements are suggested for constraining the uncertainty.
Total abatement rate ($TA$) of APCDs installed for CPP is calculated at 57% in the
provincial inventory, 6.7 % and 8.2% smaller than that in THU and AMAP/UNEP
inventories, respectively, and 12% larger than that in NJU inventory. The differences
result mainly from the varied removal efficiencies ($RE$) and application ratios ($AR$), as
shown in Table S5. For $RE$, local tests on FF, ESP+FGD and SCR+ESP+FGD were
conducted by JSEMC (2013) and Xie and Yi (2014), and the results (provided in
Table S2) are applied in the provincial inventory. From investigation on individual
plants, the $AR$ of FGD systems with relatively large benefits on Hg removal was
underestimated in NJU and overestimated in THU inventory. In the AMAP/UNEP
inventory, relevant parameters were obtained from national comment, and elevated $TA$
was estimated due to the larger $AR$ of FF and FGD and the higher $RE$ of FGD+ESP
compared to those obtained from detailed source investigation in the provincial
inventory.



For OIB, the comparison of *HgC* is similar to that for CPP. *AL* from PSC in provincial inventory is very close to that in THU inventory obtained from NSB (2011b), while *AL* in NJU inventory was much lower as the coal consumption of CEM and ISP were excluded. The *RR* from industrial boilers in this work is estimated at 82% based on domestic measurements (Wang et al., 2000; Tang et al., 2004), much lower than the result in THU inventory measured by Zhang et al. (2012), i.e., 95% for stoker fired boiler. Given the limited samples in both inventories, large uncertainty exists in *RR* of industrial boilers. Compared to the provincial inventory, *ARs* of ESP and FGD were clearly underestimated in NJU and THU inventories (Table S5), hence the *TA* in NJU was calculated 23% smaller than that in provincial inventory, leading to a 747 kg increase in Hg emission estimate. In THU inventory, however, the much higher RE of WET reduced the difference between national and provincial inventories, and *TA* in THU inventory was only 2% smaller than the provincial one.

For CEM, both the provincial and THU inventories adopted the data from Yang (2014), who measured provincial Hg contents in raw materials (limestone and other raw materials) and Hg removal efficiency of DPT+DR in China. For *AL*, the limestone consumption were calculated based on the clinker and cement production of individual plants in the provincial inventory, while THU relied on cement production at provincial level, leading to 13% smaller in *AL* and 1019 kg reduction in Hg emission estimate. In addition, consumption of other raw materials for CEM were ignored in THU inventory, leading to 1223 kg smaller in emission estimate compared to the provincial inventory. According to on-site survey by Yang (2014), fly ash is 100% reused in DPT+DR, thus the technology minimizes the Hg removal by dust collectors (ESP or FF). The *AR* of DPT+DR in THU was estimated at 82% at national average level, while it reaches 89% in Jiangsu based on detailed provincial statistics (CCA, 2011). Hence the *TA* employed in THU is 25% larger than that in provincial inventory, resulting in 259 kg underestimation in Hg emissions. NJU and AMAP/UNEP inventories failed to characterize the poor control of Hg from DPT+DR. *EFs* applied in NJU came from early domestic measurements on rotary and shaft kiln (Li, 2011; Zhang, 2007), ignoring the recent penetration of DPT+DR. In AMAP/UNEP inventory, an effective Hg capture of 40% was generally assumed for China's cement plants taking only the use of ESP and FF into account. The *TA* was estimated 215% larger than that in the provincial inventory, resulting in 2253 kg reduction in Hg emission estimate. EDGAR applied uniform emission factor (UEF) of



353 0.065g/t-clinker from EEA (2009), 32% lower than the average *EF* in the provincial

354 inventory. BNU developed S-shaped curves to estimate the time-varying dynamic

355 emission factors for non-coal combustion sector, based on the assumption of a

356 gradually declining trend in *EFs* along with increased controls of APCDs. As

357 mentioned above, however, the trend was not suitable for CEM due to the penetration

358 of DPT+DR. Thus UEF of 0.02 g/t cement estimated in BNU might result in

359 underestimation in Hg emissions, e.g., 7261 kg smaller than our provincial inventory.

360  For ISP, difficulty exists in emission estimation due to various Hg input sources

361 and complex production processes, and there is no consistent method in multi-scale

362 inventories so far. It was found that raw material production (limestone and dolomite),

363 coking, sintering and pig iron smelting with blast furnace account for most Hg

364 emissions in typical ISP in China (Wang et al., 2016). In our study, 11 factories

365 containing those processes are collected in PSC, and the emissions factors of 0.043

366 and 0.068 g/t-crude steel from Wang et al. (2016) are applied to plants with and

367 without raw material production, respectively. In other inventories, very few results

368 from domestic measurements were applied for Hg emission estimation for ISP in

369 China. NJU inventory took only coal combustion into account, and thus

370 underestimated the emissions for the sector by neglecting the Hg input along with iron

371 ore, limestone and other raw materials. THU inventory applied the emission factor of

372 0.04 g/t from Pacyna et al. (2010) for crude steel production. Besides difference in

373 emission factors, THU did not count the pig iron production in AL estimation, thus AL

374 in THU inventory is 29% lower than that in the provincial inventory, resulting in 1615

375 kg reduction in Hg emission estimate. Average *EF* in AMAP/UNEP was estimated at

376 0.039 g/t-pig iron by combining the input factor (0.05g/t-pig iron) calculated with a

377 mass balance method (UNEP, 2011a; BREF, 2012), and the removal effects of APCDs.

378 For comparison, *EF* used in our provincial inventory was recalculated at 0.064 g/t-pig

379 iron based on the hot metal charging ratio (*R* in Eq. (5); BREF, 2012). Lower *EF* in

380 AMAP/UNEP can partly be attributed to the overestimated *AR* of APCDs in ISP

381 without considering the gradual penetration of dust recycling as in CEM.

382  In general, the detailed activity and technology information including

383 manufacturing procedures and APCDs were investigated for individual plants in our

384 provincial inventory to improve the emission estimation, in contrast to previous

385 inventories that applied simplified or regional-average data. However, some crucial

386 parameters, e.g., Hg contents in coal and limestone, and Hg removal efficiencies of





APCDs, are still unavailable at plant level due to lack of measurements. Such
limitation thus indicates the necessity of more efforts on plant-specific emission
factors, and also motivates the uncertainty analysis for the provincial inventory, as
presented in Section 3.4.
**3.1.3 Comparisons of emissions for Categories 2 and 3**
For Categories 2 and 3, differences also exist in *EF* and *AL* between inventories.
For example, an emission factor of 0.22 g/t-waste combusted for MSWI based on
domestic tests (L. Chen et al., 2013; Hu et al., 2012) is applied in the provincial
inventory, while THU inventory applied 0.5 g/t from UNEP (2005), resulting in a
difference of 1024 kg in emission estimate. For primary Cu production, the provincial
inventory applied the emission factor of 0.4g/t-Cu from Wu et al. (2012), who
incorporated the results of available field measurements and the penetrations of
different smelting processes in China. BNU inventory, however, applied a much
higher emission factor at 8.9 g/t-Cu estimated by using an S-shaped curve based on
international results (Habashi, 1978; Nriagu, 1979; Pacyna, 1984; Pacyna and Pacyna,
2001; Streets et al., 2011; EEA, 2013). In NJU inventory, the emissions from NMS
and IUS were estimated much higher than the provincial inventory, attributed largely
to the different sources of activity data. For NMS, activity levels in NJU and
provincial inventories study were obtained from NSB (2011c) and NMIA (2011),
respectively. While NMIA (2011) provides the information on the production of
primary nonferrous metal (the major source of Hg emissions for NMS), the secondary
production were included in NSB (2011c), leading to possible overestimate in AL and
thereby Hg emissions. For IUS, provincial Hg consumption was allocated from the
national total use weighted by GDP in NJU inventory, while the data are directly
derived for Jiangsu from internal industrial report in the provincial inventory. In the
global inventories, moreover, all the emissions for Categories 2 and 3 in Jiangsu were
downscaled from national estimations attributed to lack of provincial information, and
big bias could be generated. For example, the large discrepancy for intentional use
and product waste associated sources between downscaled global and provincial
inventories is likely attributed to the overestimation in emissions from artisanal and
small-scale gold mining (ASGM) by global inventory (not included in Table 1 as no
ASGM was found by local source investigation).

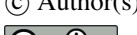



### 3.2 Hg speciation analysis of multi-scale inventories

Besides the total emissions, Hg speciation has a significant impact on the distance of Hg transport and chemical behaviors. Table 2 summarizes the mass fractions of Hg species in emissions by sector for multi-scale inventories.

In general, as shown in Table 2, reduced $Hg^0$ but enhanced $Hg^{2+}$ is estimated as the spatial scale gets smaller. This can be mainly explained by the use of domestic measurement results on Hg speciation for CEM, ISP and MSWI in the provincial inventory. For CEM, the $Hg^{2+}$ mass fraction for the dominating DPT+DR technology tends to reach 75% based on available measurements (Yang, 2014), leading to a much larger fraction of $Hg^{2+}$ emissions in the provincial inventory. In contrast, speciated Hg emissions were calculated using the same speciation profiles as those for coal combustion in NJU inventory or the uniform profile ignoring the effects of APCDs in AMAP/UNEP inventory. For ISP, heterogeneous Hg oxidation can be enhanced by the high concentration of dust and existence of $Fe_2O_3$ in the flue gas during sintering process, leading to large $Hg^{2+}$ fraction for the sector reaching 66% (Wang et al., 2016). For MSWI, results of domestic measurements (L. Chen et al., 2013; Hu et al., 2012) were applied in the provincial and NJU inventories, elevating the $Hg^{2+}$ fraction compared to THU and AMAP/UNEP inventories that applied a global uniform speciation profile without consideration of regional difference. It should be noted, however, that uncertainty exists in the estimation of speciated emissions at small spatial scale, attributed mainly to the limited samples in domestic measurements on CEM and ISP.

As mentioned above, the "universal" profiles were applied for many sectors in AMAP/UNEP inventory, ignoring the effects of various types of APCDs on Hg speciation, particularly for coal combustion. However, the fate of Hg released to atmosphere can primarily be affected by the removal mechanisms of APCDs. As shown in Table 3, for example, $Hg^0$ mass fractions for ESP+FGD and FF+FGD tend to be high reaching 83% and 78%, respectively, attributed to the relatively strong removal effects of APCDs on $Hg^{2+}$ and $Hg^p$. Once SCR is applied, an increase of $Hg^{2+}$ fraction can be observed, as the catalyst in SCR system can accelerate the conversion of $Hg^0$ to $Hg^{2+}$ (Wang et al., 2010). In addition, $Hg^0$ can also be oxidized to $Hg^{2+}$ in FF attributed to specific chemical composition in flue gas (chlorine, for example) and high temperature (Wang et al., 2008; He et al., 2012). In contrast to global inventories, therefore, national and provincial inventories take the effects of different APCDs into


account. Summarized in Table 3, considerable differences exist in the speciation
profiles for typical APCDs between national and provincial inventories, attributed
mainly to the various data used from domestic field measurements. Excluding the
measurement results on WET (Zhang et al., 2012), for example, NJU inventory
assumed the species profile from WET to be the same as CYC, and thereby largely
underestimated the mass fraction of $Hg^0$ for OIB where WET is widely applied.
Besides, the penetrations of APCDs are also crucial in determination of speciated Hg
emissions. As indicated in Table 3, with similar speciation profiles for FGD applied
between multi-scale inventories, the difference in Hg speciation is relatively small for
CPP between inventories, given the relatively accurate and transparent information on
FGD penetration in CPP used in all the inventories. For OIB, however, the difference
in Hg speciation is significantly elevated, as large diversity in APCDs penetration is
found between multi-scale inventories, as shown in Table S5. With the penetration of
FF and ESP highly underestimated, for example, THU provided a lower estimation in
$Hg^{2+}$ fraction compared to other inventories.

**3.3 Comparisons of spatial patterns of emissions between multi-scale inventories**
Figure 3 presents the spatial distributions of total and speciated Hg emissions in
Jiangsu province at 0.05˚×0.05˚. Similar patterns are found between species.
Relatively high emissions are distributed over northwestern and southern Jiangsu,
resulting from intensive coal combustion, and cement and iron & steel production, as
indicated in Figure S1 in the supplement. As an important energy base, Xuzhou in
northwestern Jiangsu contains a large number of coal combustion sources, while great
energy demand exists in southern Jiangsu attributed to highly developed economy.
The gross industrial production of the five cities in southern Jiangsu (Nanjing,
Zhenjiang, Suzhou, Wuxi and Changzhou) in 2010 accounted for 64% of the total
amount in the province. For cement production, as an example, the clinker
manufacture plants that dominate the Hg emissions compared to the subsequent
mixing stage (UNEP, 2011a), are mainly located in southern Jiangsu, depending on
the distribution of limestone resources.
In order to compare the spatial distribution of provincial inventory to that of NJU,
THU, AMAP/UNEP and EDGARv4.tox2 inventories, we upscale the gridded
provincial emissions from 0.05˚×0.05˚ to the resolutions of 0.125˚×0.125˚, 36×36km,
0.5˚×0.5˚ and 0.1˚×0.1˚ respectively. Differences in gridded $Hg^T$ emissions for




Jiangsu between the upscaled provincial inventory and other multi-scale inventories
are presented in Figure 4. Although selected sources were identified as point sources
in global/national inventories, e.g., CEM in NJU and THU, ISP in EDGARv4.tox2,
and CPP in all the inventories, the emission fraction of point sources (Categories 1
and 2) is significantly elevated to 92% in the provincial inventory. In particular, the
emissions from point sources of which the geographic information were corrected
account for 78% of total emissions in the province.
As illustrated in Figure 4, differences in gridded emissions between provincial
and other inventories NJU, THU, AMAP/UNEP and EDGARv4.tox2 are respectively
in the ranges of -760~+4135 kg, -1429~+3217 kg, -1424~+3043 kg and -1078~+3895
kg. Grids with differences more than 400 kg/yr are commonly distributed in southern
and northwestern Jiangsu, and coincide well with the locations of point sources that
are estimated to have relatively large emissions in the provincial inventory. It can thus
be indicated that differences in spatial patterns of Hg emissions come mainly from the
inconsistent information of big point sources between the provincial inventory and
national/global inventories. For CPP, AMAP/UNEP obtained information of identified
facilities                          from                          Wikipedia
(http://en.wikipedia.org/wiki/List_of_power_stations_in_Asia), and failed to include a
number of coal-fired power plants built in recent years (Steenhuisen et al., 2015). For
EDGARv4.tox2, proxy data (e.g., electricity production) from Carbon Monitoring
Action (CARMA, http://carma.org/blog/carma-notes-future-data/) are used to allocate
Hg emissions. Although CARMA incorporates all the major disclosure databases,
uncertainties still exist in certain individual plants attributed to lack of information on
geographical locations and control technologies. Moreover, as the most updated
information in CARMA was collected in 2009, EDGAR had to predict the emissions
of CPP for 2010, and thus could not fully track the actual changes in the sector, e.g.,
operation of new-built units, or shutting down the small ones. Similarly, NJU and
THU obtained the information of power units from a relatively old database (Zhao et
al., 2008), and made further assumptions on activities and penetrations of APCDs to
update the emissions of individual plants. As a result, in general, larger emissions are
found in the provincial inventory than other inventories in southern Jiangsu where big
power plants are located, particularly in Nanjing and northern Suzhou. As detailed
information at plant level is unavailable for each inventory, we speculate the
discrepancy resulted mainly from the underestimation (or missing) in coal



consumption in previous electric power generation databases that other inventories
relied on, and the use of regional/national-average information on APCD penetration
by certain inventories (e.g., THU and AMAP/UNEP). The comparison in
northwestern Jiangsu is less conclusive: the emissions in the areas with big power
plants were estimated lower in provincial inventory than AMAP/UNEP (Figure 4(c)).
Such difference, however, result not only from the varied estimations in CPP
emissions but also from discrepancy in other sources, e.g., intensive emissions from
industrial sources in the area in AMAP/UNEP. For ISP and CEM, similarly, higher
emissions were estimated by the provincial inventory for areas with big plants in
Zhenjiang, Suzhou and Changzhou in southern Jiangsu. In the provincial inventory, as
described in Section 2, the activities for each manufacturing processes were
investigated for individual plants and the information is taken into consideration in
emission estimation. In contrast, the emissions were allocated based only on the
production of individual plants in national inventories (THU and NJU), thus the
effects of manufacturing technologies on emissions were ignored. Moreover, some
CEM and ISP plants were missed in those national inventories, leading to
underestimation in emissions for corresponding regions. In general, due to lack of
plant-specific information, previous inventories failed to capture the relatively large
emissions from big point sources. When the national inventory was applied in CTM,
the simulated concentrations of $Hg^T$ were usually lower than the observation at rural
sites in eastern China (Wang et al., 2014). Since many big plants are commonly being
moved from urban to rural areas (Zhao et al., 2015b; Zhou et al., in preparation),
improvement in model performance could be expected when the elevated emissions in
rural areas are estimated and used for CTM, incorporating the accurate information of
individual big plants.

With much fewer big emitters, discrepancies in gridded emissions for other part

of Jiangsu resulted largely from the allocation of considerable emissions as area
sources in national and global inventories. For example, in spite of an estimation of
8496 kg smaller than the provincial inventory in total emissions, NJU inventory
applied proxies (e.g., population and GDP) to allocate the emissions except those
from CPP, resulting in higher emissions in central and most part of northern Jiangsu
(Figure 4(a)). Similar patterns are also found for THU (Figure 4(b)) and
AMAP/UNEP (Figure 4(c)) compared to provincial inventory.

Besides the total emissions, differences in spatial distribution of speciated Hg





emissions between multi-scale inventories are presented in Figure S3 in the
supplement. The various patterns for species are largely influenced by the distribution
of different types of big point sources, as the speciation profiles vary significantly
between source types in the national and provincial inventories (Table 2). Compared
to other inventories, larger $Hg^0$ emissions were found in the provincial inventory in
southern Jiangsu (particularly Zhenjiang and Taizhou) where CPPs that have large
fraction of $Hg^0$ are intensively located. Elevated $Hg^{2+}$ emissions were dominated by
intensive CEM industry in Changzhou, Wuxi and Zhenjiang in southern Jiangsu, as
the $Hg^{2+}$ fraction of CEM reaches 73% in the provincial inventory. In contrast to $Hg^0$
and $Hg^{2+}$, differences in $Hg^P$ emissions between inventories in central Jiangsu are
closely related with the locations of OIB plants, attributed mainly to the relatively
poor understanding of the particle control and thereby $Hg^p$ release of OIB. The
emissions in the provincial inventory is larger than THU but smaller than
AMAP/UNEP, as the $Hg^p$ mass fraction of OIB was assumed at 2% in THU while it
reached 10% in AMAP/UNEP (Table 2).
The vertical distribution of Hg releases, which is crucial for the transport range
of atmospheric Hg, is also analyzed in this work. Four groups of release height are
defined: 0-58m, 58-141m, 141-250m and >250m. Based on the detailed information
of emission sources, the fractions of Hg releases into the four groups for CPP are 2%,
66%, 31%, and 1%, respectively, and the analogue numbers for OIB, ISP, and CEM
are 85%, 13%, 2%, and 0%; 4%, 44%, 12%, and 4%; and 6%, 94%, 0%, and 0%,
respectively. The release heights for rest sources are uniformly assumed at the range
of 0-58m. As a result, the fractions of total Hg emissions in the four groups are
estimated as 35%, 53%, 11% and 1%. In AMAP/UNEP inventory, as a comparison,
the fractions at the height of 0-50m, 50-150m and >150m were estimated at 23%,
53% and 24% respectively, with larger share in Hg emitted over 150m than that in our
provincial inventory.

**3.4 Uncertainty of the provincial inventory**
As summarized in Table 4, the uncertainties of speciated Hg emissions in the
provincial inventory are estimated at -24%~+82% (95% confidence intervals (CI)
around central estimates), -34%~+99%, -23%~68%, and -34%~+270% for $Hg^T$, $Hg^0$,
$Hg^{2+}$ and $Hg^p$, respectively. For comparison, the uncertainties of Jiangsu emissions
from major sectors including CPP, CEM, ISP and OIB in NJU inventory are



recalculated following Zhao et al. (2015a) and provided in Table 4 as well. As can be
seen, the uncertainties for major sources in the provincial inventory were smaller than
those in NJU inventory, attributed largely to the bottom-up approach used in
provincial inventory with more accurate information on activity levels and APCDs
applications for individual plants of Category 1. In addition, with more field
measurements on Hg contents in coal and limestone incorporated, the uncertainties of
$HgC_{raw}$ and $HgC_{Limstone}$ are significantly reduced, resulting from the mechanism of
error compensation when $HgC_{raw}$ of coals produced in different provinces are taken
into account in the inter-provincial flow model for coal transport, and the successful
application of bootstrap simulation, respectively. As a result, the uncertainties of
emissions from CPP, OIB and CEM are effectively reduced in the provincial
inventory.

The parameters contributing most to uncertainties and their contributions to the

variance of corresponding emission estimates are summarized by sector in Table S8 in
the supplement. For CPP and OIB, parameters related to emission factors contribute
most to the uncertainties of $Hg^T$ emissions, including the $HgC_{raw}$ in provinces with
largest contribution to the input of coal consumed in Jiangsu (i.e., Shaanxi and Inner
Mongolia), and the removal efficiencies ($RE$) or release ratios ($RR$) of Hg for typical
APCD (ESP+FGD) and combustor type (grate boiler). $HgC_{raw}$ of coals produced in
Shaanxi and Inner Mongolia that collectively accounted for 34% of coal consumption
in Jiangsu, contributed 26% and 18% to the uncertainties of Hg emissions for CPP,
and 15% and 11% to those for OIB, respectively. It is thus essential to conduct
systematic and synergetic measurements on $HgC_{raw}$ in different regions (particularly
those with large coal production) to constrain the uncertainties of Hg emission
estimation for coal combustion sources, at both regional and national scales. Given
the wide application of ESP+FGD in CPP (70% in coal consumption), $RE_{ESP+FGD}$ is
estimated to contribute 20% to Hg emissions from CPP. Local measurements on $RE$ of
typical APCDs, which have started in Jiangsu (JSEMC, 2013; Xie and Yi, 2014), are
expected to potentially improve the Hg emission estimation at regional level.
Although applied in 92% of OIB plants in Jiangsu, there are very few studies on Hg
release rate of grate boiler, resulting in a contribution of 5% to the emission
uncertainty. For CEM, $HgC_{Limestone}$ dominates the uncertainties of Hg emissions, with
the contribution estimated at 84%. Attributed to lack of detailed information,
provincial average of $HgC_{Limestone}$ with the lognormal distribution fitted through





bootstrap simulation based on available measurements (Figure S2 and Table S6) was
uniformly applied for all the individual plants, leading to the enhanced contribution to
the uncertainty. For ISP, *EF* of limestone and dolomite production contributes 60% to
Hg emissions, as the process is estimated to account for 88% of emissions from the
entire sector. In addition, *AL* from the biggest ISP factory, which accounted for 40%
and 75% of pig iron and crude steel production for the whole province, respectively,
contributes 24% to the total uncertainty of ISP sector. The result indicates a necessity
of specific investigation on super emitters. For rest sources, MSWI, BIO and O&G are
the biggest sources for $Hg^T$ emissions, and *EFs* of those types of sources thus
contribute most to the emission uncertainty.

In most cases, parameters with big contribution to uncertainty of $Hg^T$ also play

crucial roles in uncertainty of speciated emissions. Moreover, the speciation profiles
for typical source types and APCDs are identified as key parameters to the
uncertainties of speciated emissions as well. For example, the mass fraction of $Hg^{2+}$
from ESP+FGD, and that of $Hg^p$ from ESP are the biggest contributors to
uncertainties of $Hg^{2+}$ and $Hg^p$ emissions from CPP, respectively. For OIB, the mass
fraction of $Hg^p$ from sources without any control is much higher than those with
APCDs (Table 3), thus it plays an important role in the emission uncertainty, with the
contribution estimated at 35%. For CEM and ISP, studies on speciation profiles are
limited so far, and the speciation profiles for DPT+DR and ISP plants contribute
largely to uncertainties of speciated emissions.

**4 CONCLUSIONS**

Taking Jiangsu province in China as an example, the discrepancies and their

sources of atmospheric Hg emission estimations in multi-scale inventories applying
varied methods and data are thoroughly analyzed. Using a bottom-up approach that
integrates best available information of individual plants and most recent field
measurements, the total Hg emissions in Jiangsu 2010 are calculated at 39 105 kg, and
the estimate is larger than any other national/global inventories. CPP, ISP, CEM and
OIB collectively accounted for 90% of the total emissions. Comparisons between
available studies demonstrate that the information gaps of multi-scale inventories lead
to big differences in Hg emission estimation. Discrepancies in emissions between
inventories for the above-mentioned major sources come primarily from various data



sources for activity levels, Hg contents in coals and total abatement effects of APCDs.
Notable increase in $Hg^{2+}$ emissions is estimated with the bottom-up approach
compared to other global/national inventories, attributed mainly to the adoption of
domestic measurement results with elevated mass fraction of $Hg^{2+}$ for CEM, ISP and
MSWI. Inconsistent information of big point sources lead to large differences in
spatial distribution of emissions between provincial and other inventories, particularly
in southern and northwestern of the province where intensive coal combustion and
industry are located. Improved estimates in emission level, speciation and spatial
distribution are expected to better support the regional chemistry transport modeling
of atmospheric Hg. Compared to the national inventory, uncertainties of Hg emissions
are reduced in provincial inventory using the bottom-up approach. Extensive and
dedicated measurements are urgently suggested on Hg contents in coal/limestone and
removal efficiency of dominating APCDs to further improve the emission estimation
at regional/local scales.

**ACKNOWLEDGEMENT**
This work was sponsored by the Natural Science Foundation of China
(41575142), Natural Science Foundation of Jiangsu (BK20140020), Jiangsu Science
and Technology Support Program (SBE2014070918), and Special Research Program
of Environmental Protection for Commonweal (201509004). We would like to
acknowledge Hezhong Tian from Beijing Normal University and Simon Wilson from
UNEP/AMAP Expert Group for the detailed information on national/global Hg
emission inventories.

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



**TABLES**

**Table 1 Emission estimates for Jiangsu in 2010 and species from multi-scale inventories by sector. Recall from Section 2 the abbreviations for emission sources: CPP: coal-fired power plants; RCC: residential coal combustion; O&G: oil and gas combustion; OIB: other industrial coal combustion; CEM: cement production; ISP: iron & steel plants; NMS: nonferrous metal smelting; AP: aluminum production; LGM: large-scale gold mining; MM: mercury mining; HC: human cremation; MSWI: municipal solid waste incineration; RSWI: rural solid waste incineration; BFLP: battery/fluorescent lamp production; BIO: biofuel use/biomass open burning; and PVC: PVC production.**

| | | CPP[1] | RCC[3] | O&G[3] | OIB[1] | CEM[1] | ISP[1] | NMS[2] | AP[2] | LGM | MM | HC[3] | MSWI[2] | RSWI[3] | BFLP[2] | BIO[3] | PVC[2] | Total |
|---|---|---|---|---|---|---|---|---|---|---|---|---|---|---|---|---|---|---|
| $Hg^{T}$ | Bottom-up | 11549 | 195 | 930 | 8652 | 9264 | 5654 | 91 | 29 | 0 | 0 | 326 | 1009 | 365 | 158 | 461 | 423 | 39106 |
| | NIU | 11208 | 165 | 930 | 6901 | 1137 | 2243 | 2158 | / | 23 | 51 | / | 1009 | 457 | 1225 | 500 | 2603 | 30610 |
| | THU | 10768 | 345 | 752 | 10680 | 8238 | 2539 | 0 | 29 | 0 | 0 | 326 | 2294 | 244 | / | 219 | / | 36434 |
| | BNU | 12883 | 267 | 898 | 10172 | 3288 | 2669 | 2022 | 6 | / | / | / | 308 | | / | 303 | / | 32816 |
| | AMAP/UNEP | | 9292[a] | / | | | | 17759[b] | | | | | | 4976[c] | | | | 32027 |
| | EDGARv4.tox2 | 10233[d] | 1181[e] | | 3310[f] | 6364 | 447 | 413 | | | | | 1017 | | | 43 | | 23008 |
| $Hg^{0}$ | Bottom-up | 8811 | 91 | 465 | 4689 | 2461 | 1908 | 45 | 23 | 0 | 0 | 313 | 1017 | 47 | 158 | 350 | 423 | 20801 |
| | NIU | 8133 | 42 | 465 | 2042 | 685 | 1208 | 1189 | / | 16 | 41 | / | 190 | | 980 | 380 | 2082 | 17453 |
| | THU | 7689 | 247 | 376 | 6995 | 2793 | 863 | 0 | 23 | 0 | 0 | 313 | 2202 | 244 | / | 162 | / | 21907 |
| | AMAP/UNEP | | 4646[a] | | | | | 14207[b] | | | | | | 4668[c] | | | | 23521 |
| $Hg^{2+}$ | Bottom-up | 2653 | 73 | 372 | 3394 | 6752 | 3746 | 45 | 4 | 0 | 0 | 0 | 868 | 314 | 0 | 23 | 0 | 18244 |
| | NIU | 2900 | 45 | 372 | 4003 | 431 | 835 | 963 | / | 7 | 8 | / | 868 | 59 | 184 | 25 | 390 | 11090 |
| | THU | 3058 | 92 | 301 | 3471 | 5338 | 1676 | 0 | 4 | 0 | 0 | 0 | 0 | 0 | 0 | 11 | / | 13951 |
| | AMAP/UNEP | | 3717[a] | | | | | 2707[b] | | | | | | 238[c] | | | | 6662 |
| $Hg^{p}$ | Bottom-up | 85 | 32 | 93 | 569 | 51 | 0 | 0 | 1 | 0 | 0 | 13 | 10 | 4 | 0 | 88 | 0 | 946 |
| | NIU | 175 | 79 | 93 | 855 | 20 | 200 | 6 | / | 0 | 3 | / | 10 | 5 | 61 | 95 | 130 | 1732 |
| | THU | 22 | 6 | 75 | 214 | 107 | 0 | 0 | 1 | 0 | 0 | 13 | 92 | 0 | 0 | 46 | / | 576 |
| | AMAP/UNEP | | 929[a] | | | | | 845[b] | | | | | | 70[c] | | | | 1844 |

[1, 2, 3] Sectors in category 1, 2 and 3 as classified in Section 2. [a] Stationary combustion sources: power plants, distributed heating, and other energy use (industrial sources excluded). [b] Industrial sources including stationary combustion for industry, CEM, ISP, NMS, AP, LGM and MM. [c] Intentional use and product waste associated sources: artisanal and small-scale gold mining, solid waste incineration and other product waste disposal, chlor-alkali industry, and human cremations. [d, e, f] Both coal and other fossil fuel combustion included.





**Table 2 Hg speciation profiles by sector and the mass fractions to total emissions in multi-scale inventories (%).**

| Sector | Provincial inventory | | | NJU | | | THU | | | AMAP/UNEP | | |
|---|---|---|---|---|---|---|---|---|---|---|---|---|
| | $Hg^0$ | $Hg^{2+}$ | $Hg^p$ | $Hg^0$ | $Hg^{2+}$ | $Hg^p$ | $Hg^0$ | $Hg^{2+}$ | $Hg^p$ | $Hg^0$ | $Hg^{2+}$ | $Hg^p$ |
| CPP | 76 | 23 | 1 | 73 | 26 | 2 | 71 | 28 | 0 | 50 | 40 | 10 |
| RCC | 46 | 37 | 16 | 25 | 27 | 48 | 71 | 27 | 2 | 50 | 40 | 10 |
| O&G | 50 | 40 | 10 | 50 | 40 | 10 | 50 | 40 | 10 | 50 | 40 | 10 |
| OIB | 54 | 39 | 7 | 30 | 57 | 13 | 66 | 33 | 2 | 50 | 40 | 10 |
| CEM | 27 | 73 | 1 | 60 | 38 | 2 | 34 | 65 | 1 | 80 | 15 | 5 |
| ISP | 34 | 66 | 0 | 54 | 37 | 9 | 34 | 66 | 0 | 80 | 15 | 5 |
| NMS | 50 | 50 | 0 | 55 | 45 | 0 | | / | | 80 | 15 | 5 |
| AP | 80 | 15 | 5 | | / | | 80 | 15 | 5 | 80 | 15 | 5 |
| LGM | | | | 70 | 30 | 0 | | | | 80 | 15 | 5 |
| MM | | / | | 80 | 15 | 5 | | / | | 80 | 20 | 0 |
| ASGM | | | | | / | | | | | 100 | 0 | 0 |
| HC | 96 | 0 | 4 | | | | 96 | 0 | 4 | 80 | 15 | 5 |
| MSWI | 13 | 86 | 1 | 13 | 86 | 1 | 96 | 0 | 4 | 20 | 60 | 20 |
| BFLP | 100 | 0 | 0 | 80 | 15 | 5 | 100 | 0 | 0 | 80 | 15 | 5 |
| BIO | 76 | 5 | 19 | 76 | 5 | 19 | 74 | 5 | 21 | | / | |
| PVC | 100 | 0 | 0 | 80 | 15 | 5 | | / | | | | |
| Total | 51 | 47 | 2 | 57 | 37 | 6 | 60 | 38 | 2 | 73 | 21 | 6 |



**Table 3 Hg speciation profiles used in provincial and national inventories for typical APCDs (%).**

| Sources | | Hg speciation | | | | | | | | |
|---|---|---|---|---|---|---|---|---|---|---|
| | | Provincial inventory | | | NJU | | | THU | | |
| | | $Hg^0$ | $Hg^{2+}$ | $Hg^p$ | $Hg^0$ | $Hg^{2+}$ | $Hg^p$ | $Hg^0$ | $Hg^{2+}$ | $Hg^p$ |
| Coal combustion | ESP | 57 | 41 | 1 | 65 | 35 | 0 | 58 | 41 | 1 |
| | FF | 31 | 61 | 7 | 16 | 73 | 11 | 50 | 49 | 1 |
| | WET | 65 | 33 | 2 | 30 | 57 | 13 | 65 | 33 | 2 |
| | CYC | 30 | 57 | 14 | 30 | 57 | 13 | | / | |
| | ESP+FGD | 83 | 16 | 0 | 83 | 16 | 0 | 84 | 16 | 1 |
| | SCR+ESP+FGD | 71 | 29 | 0 | 72 | 28 | 0 | 74 | 26 | 0 |
| | FF+FGD | 78 | 21 | 1 | | / | | 78 | 21 | 1 |
| | No | 48 | 34 | 18 | 24 | 20 | 56 | 56 | 34 | 10 |
| CEM | DPT+DR/FF* | 24 | 75 | 1 | 16 | 73 | 11 | 24 | 76 | 1 |
| | SKT/ESP* | 83 | 16 | 1 | 65 | 35 | 0 | 80 | 15 | 5 |
| | RKT/WET* | 47 | 51 | 1 | 30 | 57 | 14 | 80 | 15 | 5 |

*: DPT+DR, SKT and RKT for provincial and THU inventory (Zhang et al., 2015); FF, ESP and WET for NJU inventory (Zhao et al., 2015).





**Table 4. Uncertainties of Hg emissions in Jiangsu in provincial and national (NJU) inventories by source, expressed as the 95% confidence intervals of central estimates.**

|  | Sources | $Hg^T$ | $Hg^0$ | $Hg^{2+}$ | $Hg^p$ |
|---|---|---|---|---|---|
| Provincial | CPP | (-59%, +147%) | (-64%, +131%) | (-56%, +244%) | (-43%, +418%) |
|  | CEM | (-15%, +58%) | (-36%, +87%) | (-18%, +63%) | (-57%, +218%) |
|  | ISP | (-38%, +53%) | (-33%, +156%) | (-62%, +44%) | / |
|  | OIB | (-52%, +138%) | (-55%, +133%) | (-55%, +146%) | (-67%, +329%) |
|  | Rest sources | (-25%, +133%) | (-20%, +151%) | (-67%, +168%) | (-43%, +367%) |
|  | Total | (-26%, +81%) | (-34%, +99%) | (-23%, +68%) | (-34%, +270%) |
| NJU | CPP | (-80%, +198%) | (-80%, +198%) | (-80%, +201%) | (-75%, +477%) |
|  | CEM | (-62%, +97%) | (-75%, +140%) | (-63%, +82%) | (-73%, +266%) |
|  | ISP | (-81%, +167%) | (-82%, +157%) | (-82%, +170%) | (-81%, +250%) |
|  | OIB | (-83%, +153%) | (-97%, +218%) | (-97%, +228%) | (-87%, +170%) |





# FIGURES

**Fig. 1. The ratios of estimated Hg emissions for Jiangsu 2010 in global/national inventories to that in provincial inventory for selected sources and anthropogenic total.**

**Fig. 2. Sensitivity analysis of selected parameters in Hg emission estimation for Category 1 sources. (a) Relative changes in parameters, calculated using Eq. (6); (b) Changes in emissions when parameters in the provincial inventory were replaced with those in other inventories, calculated using Eq. (7). $HgC_{raw}$: Hg content in raw coal; AL: activity levels as raw coal consumption by CPP and OIB, limestone used by CEM, and crude steel produced in ISP; TA: total abatement rate of APCDs; RR: Hg release rate for combustion; IEF: input emission factors (before control of APCDs); UEF: uniform emission factor (without consideration of different APCD types); $EF_{iron}$ and $EF_{steel}$: emission factors of pig-iron and steel production respectively.**

**Fig. 3. Spatial distribution of Hg emissions for Jiangsu 2010 at a resolution of $0.05^o \times 0.05^o$ for (a) $Hg^T$, (b) $Hg^0$, (c) $Hg^{2+}$, and (d) $Hg^p$.**

**Fig. 4. Differences in gridded $Hg^T$ emissions in Jiangsu 2010 between provincial and other inventories: emissions in provincial inventory minus those in NJU (a), THU (b), AMAP/UNEP (c) and EDGARv4.tox2 (d). The locations of point sources with relatively large Hg emissions estimated in provincial inventory are indicated in the panels as well.**





**Fig. 1**

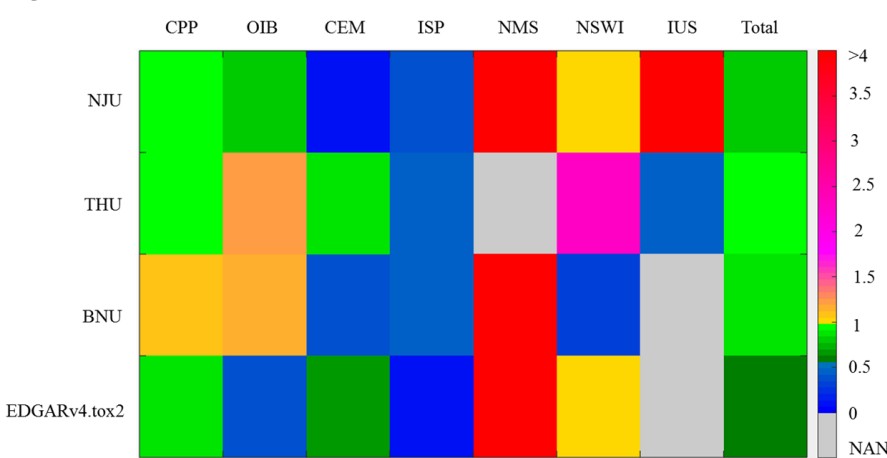





**Fig. 2.**







**Fig. 3.**

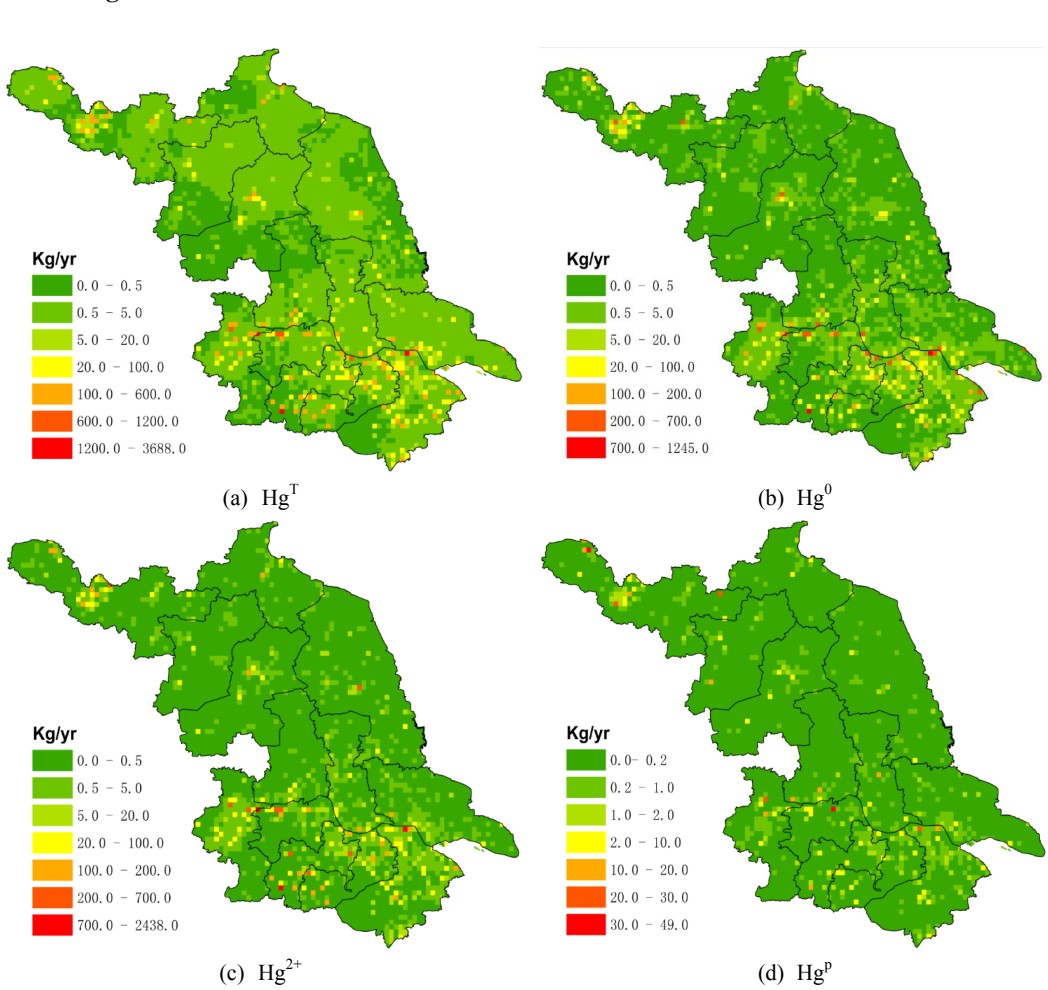





**Fig. 4.**

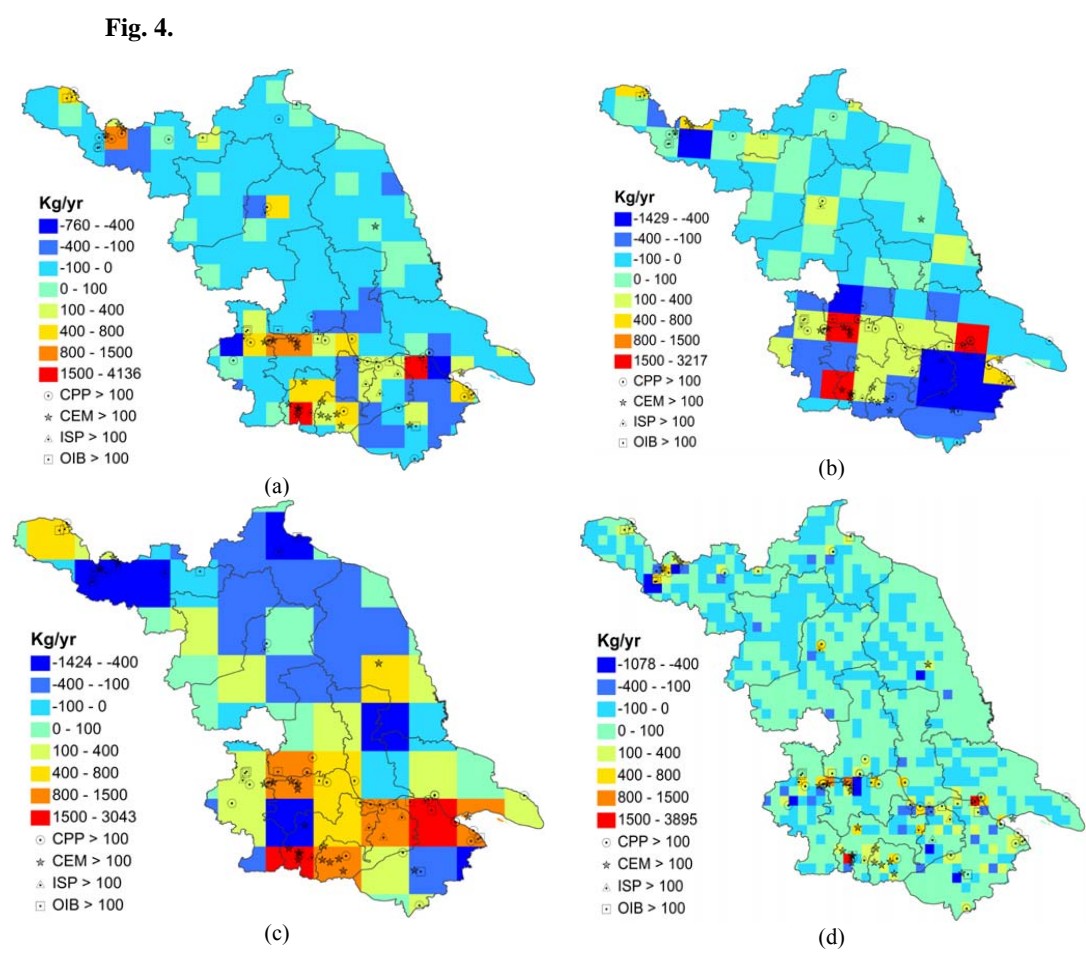

(a)

(b)

(c)

(d)