# Peer review of "A high-resolution regional emission inventory of atmospheric mercury and its comparison with multi-scale inventories: a case study of Jiangsu, China"

_Atmospheric Chemistry and Physics, 2016_

## Referee Comment (RC1) · Anonymous Referee #2 · 19 Aug 2016

In this study, the authors developed a high-resolution Hg emission inventory of anthropogenic origin for 2010. The provincial inventory was compared to selected global and national inventories. Discrepancies in emission levels, speciation, and spatial distributions are evaluated. The major contribution of the study is comparison of the inventories, and identifying the effects of different approaches and data on developing the inventories.

The study is relevant since there are considerable information gaps between multiscale inventories. The differences attribute mainly to the data of different sources and

levels of details. A bottom-up approach used in this study could help improve the precision of the inventory.

A key question is, the authors indicated that part of the data are internal data from Environmental Protection Agency of Jiangsu Province, and the internal industry reports. We would like to see more explanations on these "internal data".

Line 78, "there are currently very few studies focusing on Hg at regional/local scales". This is not true.

Line 128-131, could you provide more information on the PSC? Any difference between PSC and published statistical data?

Line 180, please explain the internal industry reports.

It would be interesting if at the end of the manuscript, the authors might give some discussions on the possibility of overall underestimation of mercury inventory for China, not just for the province. That is to say, the same problems in other national inventories might happen in other provinces in China.

---

## Referee Comment (RC2) · Anonymous Referee #1 · 23 Aug 2016

The article presents a comparison of international, national and a new local bottom-up Hg emission inventory for the Jiangsu region in China. The study highlights the serious discrepancies, in both emission totals and speciation, between emission inventory estimates. This has serious implications for the regional atmospheric Hg burden and deposition flux. If the underestimate for Jiangsu is representative for the major economies of the region then this would have global repercussions. Unfortunately the authors do not comment on how wide-spread the underestimations in Hg emissions they have identified for Jiangsu may be. Are the shortcomings in the national and global inventories identified for Jiangsu applicable to other heavily industrialised regions of China? It

would improve the article if the authors could provide estimates of the possible range of underestimation of Chinese emissions and how this would influence the global Hg emissions total. The difference in Hg emission speciation (and to a lesser extent emission height) between the inventories will have an impact on local deposition and Hg export estimates from the region, neither of these aspects are discussed in any detail.

The description of the database compilation is thorough but rather repetitive of previous work. The English requires substabtial improvement and overall the manuscript could be more concise.

Collaboration with modelling groups or at least performing some trajectory calculations with the previous and revised speciation would make the paper far more interesting. Making the emissions database available would seem a good idea as I am sure it would lead to fruitful joint research beneficial not only to the science community but also to local environmental agencies and policy makers. The fact that some of the data sources are not publicly available is a concern.

However the evidence presented raises important questions concerning the accuracy of current emission inventories, and in particular global inventories and warrants publication.

Sections 2.1 and 2.2 could be shortened with reference to sections 2.1 and 2.3 of Zhao et al., 2015 (Evaluating the effects of China's pollution controls on inter-annual trends and uncertainties of atmospheric mercury emissions, Atmos. Chem. Phys., 15, 4317-4337), which are very similar. Section 2.3, is this really a sensitivity analysis, or more simply an analysis of the scale of the differences in emissions which result from the assumptions made in the compilation of the inventories? Section 3.1.2 particularly is rather long and full of acronyms, it would likely aid the reader if it were divided into subsections. Section 3.3 would also benefit from being more concise.

---

## Author Comment (AC1) · 2 Nov 2016

Main revisions and response to reviewers' comments

Manuscript No.: acp-2016-540

Title: A high-resolution regional emission inventory of atmospheric mercury and its comparison with multi-scale inventories: a case study of Jiangsu, China

Authors: Hui Zhong, Yu Zhao, Marilena Muntean, Lei Zhang, Jie Zhang

We thank very much for the valuable comments from the reviewer, which help us improve the quality of our manuscript. The comments were carefully considered and revisions have been made in response to the comments and suggestion. The major revisions were marked in red bold in the submitted manuscript. Our responses to each comment or suggestion are provided in details as below, along with the brief description on the revision actions taken in the revised manuscript.

In this study, the authors developed a high-resolution Hg emission inventory of anthropogenic origin for 2010. The provincial inventory was compared to selected global and national inventories. Discrepancies in emission levels, speciation, and spatial distributions are evaluated. The major contribution of the study is comparison of the inventories, and identifying the effects of different approaches and data on developing the inventories. The study is relevant since there are considerable information gaps between multi-scale inventories. The differences attribute mainly to the data of different sources and levels of details. A bottom-up approach used in this study could help improve the precision of the inventory.

Response and revisions:

We appreciate the reviewer's positive remarks.

Q1. A key question is, the authors indicated that part of the data are internal data from Environmental Protection Agency of Jiangsu Province, and the internal industry reports. We would like to see more explanations on these "internal data". (Line 128-131, could you provide more information on the PSC? Any difference between PSC and published statistical data? Line 180, please explain the internal industry reports.)

Response and revisions:

Pollution Source Census (PSC) was conducted by local environmental protection agencies, in which the data for individual emission sources were collected and compiled through on-site investigation, including manufacturing technology, production level, energy consumption, fuel quality, and emission control device. Compared to the energy

and economic statistics at sector level that were commonly used in global/national inventories, we believe the plant-by-plant PSC data could provide more detailed and accurate information on individual emission sources, particularly for power and industrial plants. Moreover, differences in total energy consumption and industrial production levels exist between the PSC data and the energy/economic statistics. For example, the coal consumption by CPP in PSC for Jiangsu 2010 was 6% larger than the provincial statistics.

Internal industry reports indicate the association commercial reports that provide the activity data of intentional Hg use. Rarely included in the national or provincial statistics, the data were collected at http://www.askci.com/.

We have included the information in lines 128-138, Pages 5-6 and in lines 187-188, Page 7 in the revised manuscript, respectively.

Q2. Line 78, "there are currently very few studies focusing on Hg at regional/local scales". This is not true.

Response and revisions:

We thank the reviewer's reminder. The sentence was revised as "there are currently very few studies on Hg emissions at regional/local scales in China", in lines 77-78, Page 4 in the revised manuscript.

Q3. It would be interesting if at the end of the manuscript, the authors might give some discussions on the possibility of overall underestimation of mercury inventory for China, not just for the province. That is to say, the same problems in other national inventories might happen in other provinces in China.

Response and revisions:

We thank the reviewer's important comment, and it is similar to Q1 from another reviewer. Through the comparisons between provincial and other downscaled global/national inventories, it could be found that cement and iron & steel industries

were the two most important sectors of which the Hg emissions were significantly underestimated by previous inventories. The underestimations came mainly from the ignorance of high Hg release ratio of precalciner technology with dust recycling, and/or application of relatively low emission factors for steel production. For example, the estimation of CEM and ISP emissions by the national inventory (Zhao et al., 2015a) was 77% lower than the provincial one, and the difference accounted for 30% of the total anthropogenic Hg emissions from the provincial inventory. Compared to the provincial inventory, for example, we could thus cautiously infer that Hg emissions might also be underestimated for other regions with intensive cement and steel industries in China in previous inventories. For other big sources, e.g., power plants and industrial boilers, the Hg emissions were influenced largely by the Hg contents in coal and the application of emission control devices. Whether the emissions of those sources were underestimated or not for other parts of the country could hardly be judged unless detailed information gets available for the regions. In general, however, the method developed and demonstrated for Jiangsu in this work could be promoted to other provinces, particularly for those with intensive industrial plants. With the detailed data on individual sources sufficiently applied, the accuracy in China's Hg emission estimation can be expected to be largely improved.

We presented the discussions in lines 666-682, Page 22 at the end of the revised manuscript.
* * *

---

## Author Comment (AC2) · 2 Nov 2016

Main revisions and response to reviewer's comments

Manuscript No.: acp-2016-540

Title: A high-resolution regional emission inventory of atmospheric mercury and its comparison with multi-scale inventories: a case study of Jiangsu, China Authors: Hui Zhong, Yu Zhao, Marilena Muntean, Lei Zhang, Jie Zhang

We thank very much for the valuable comments from the reviewer, which help us im-

prove the quality of our manuscript. The comments were carefully considered and revisions have been made in response to the comments and suggestion. The major revisions were marked in red bold in the submitted manuscript. Our responses to each comment or suggestion are provided in details as below, along with the brief description on the revision actions taken in the revised manuscript.

The article presents a comparison of international, national and a new local bottom-up Hg emission inventory for the Jiangsu region in China. The study highlights the serious discrepancies, in both emission totals and speciation, between emission inventory estimates. This has serious implications for the regional atmospheric Hg burden and deposition flux. If the underestimate for Jiangsu is representative for the major economies of the region then this would have global repercussions.

Response and revisions:

We appreciate the reviewer's positive remarks.

Q1. Unfortunately the authors do not comment on how wide-spread the underestimations in Hg emissions they have identified for Jiangsu may be. Are the shortcomings in the national and global inventories identified for Jiangsu applicable to other heavily industrialised regions of China? It would improve the article if the authors could provide estimates of the possible range of underestimation of Chinese emissions and how this would influence the global Hg emissions total.

Response and revisions:

We thank the reviewer's important comment. Through the comparisons between provincial and other downscaled global/national inventories, it could be found that cement and iron & steel industries were the two most important sectors of which the Hg emissions were significantly underestimated by previous inventories. The underestimations came mainly from the ignorance of high Hg release ratio of precalciner technology with dust recycling, and/or application of relatively low emission factors for steel

production. For example, the estimation of CEM and ISP emissions by the national inventory (Zhao et al., 2015a) was 77% lower than the provincial one, and the difference accounted for 30% of the total anthropogenic Hg emissions from the provincial inventory. Compared to the provincial inventory, for example, we could thus cautiously infer that Hg emissions might also be underestimated for other regions with intensive cement and steel industries in China in previous inventories. For other big sources, e.g., power plants and industrial boilers, the Hg emissions were influenced largely by the Hg contents in coal and the application of emission control devices. Whether the emissions of those sources were underestimated or not for other parts of the country could hardly be judged unless detailed information gets available for the regions. In general, however, the method developed and demonstrated for Jiangsu in this work could be promoted to other provinces, particularly for those with intensive industrial plants. With the detailed data on individual sources sufficiently applied, the accuracy in China's Hg emission estimation can be expected to be largely improved.

We presented the discussions in lines 666-682, Page 22 at the end of the revised manuscript.

Q2. The difference in Hg emission speciation (and to a lesser extent emission height) between the inventories will have an impact on local deposition and Hg export estimates from the region, neither of these aspects are discussed in any detail. Response and revisions:

We thank the reviewer's comment. Relevant discussions have been added in lines 576-581, Page 19 at the end of Section 3.3 in the revised manuscript:

The smaller fraction of Hg emissions under 150m and larger fraction of Hg2+ as discussed in Section 3.2 in the provincial inventory are expected to result in more local deposition and less long-range transport compared to previous inventories when they are applied in CTM. The re-emissions of legacy Hg could then be enhanced and make a significant contribution to atmospheric Hg concentrations, as indicated by Zhu et al.

(2012).

Q3. The description of the database compilation is thorough but rather repetitive of previous work. The English requires substantial improvement and overall the manuscript could be more concise.

Response and revisions:

We thank the reviewer's comment. The description of the database compilation is given in Section 2.3, and databases for Hg emission factors/related parameters are provided in the supplement avoiding unnecessary description. We have also tried our best to shorten the manuscript and to make it more concise.

Q4. Collaboration with modelling groups or at least performing some trajectory calculations with the previous and revised speciation would make the paper far more interesting.

Response and revisions:

We thank the reviewer's important comment. We agree that chemistry transport modeling (CTM) is a very crucial step to evaluate the emission inventory, and it is exactly what we are working on. We are currently conducting the Hg simulation at provincial scale with WRF-CMAQ-Hg, using the different inventories mentioned in this paper. The improvement in revised provincial inventory is expected to be evaluated by comparing the model performances with various inventories. We hope the work could be finished and a companion paper would come out soon.

Q5. Making the emissions database available would seem a good idea as I am sure it would lead to fruitful joint research beneficial not only to the science community but also to local environmental agencies and policy makers. The fact that some of the data sources are not publicly available is a concern.

Response and revisions:
* * *
Interactive
comment

We thank the reviewer's reminder and totally agree. We will upload the data to the website of our group. The data will be available online soon at http://www.airqualitynju.com/En/Default. We have stated this at the end of the revised manuscript.

Q6. Sections 2.1 and 2.2 could be shortened with reference to Sections 2.1 and 2.3 of Zhao et al., 2015 (Evaluating the effects of China's pollution controls on inter-annual trends and uncertainties of atmospheric mercury emissions, Atmos. Chem. Phys., 15, 4317-4337), which are very similar.

Response and revisions:

We thank the reviewer's comment and have tried to shorten the sections. For example, in lines 188-189, Page 7 in the revised manuscript, we have stated:

Activity data for MSWI, RSWI and BIO are taken following Zhao et al. (2015a).

It should be noted, however, that the provincial inventory is established with a bottom-up method, which is quite different from the approach by Zhao et al. (2015a). Thus some details in the provincial inventory approach must be given to avoid confusion.

Q7. Section 2.3, is this really a sensitivity analysis, or more simply an analysis of the scale of the differences in emissions which result from the assumptions made in the compilation of the inventories?

Response and revisions:

We thank the reviewer's comment. We agree with the reviewer that the analysis here is to quantify the scale of emission changes resulting from varied values of given parameters in different inventories. In the analysis, we include both the differences in assumptions for key parameters and the scale of corresponding emission changes due to the varied assumptions. We mean the analysis can thus show the sensitivity of the emissions to specific parameter.

Q8. Section 3.1.2 particularly is rather long and full of acronyms, it would likely aid the reader if it were divided into subsections.

Response and revisions:

We thank the reviewer's comment. Now the original Section 3.1.2 was divided into two sections, Section 3.1.2 for power plants and industrial boilers, and 3.1.3 for cement and iron & steel industries.

Q9.Section 3.3 would also benefit from being more concise.

Response and revisions:

We thank the reviewer's comment and have tried our best to shorten the section.